METHODS AND RESOURCES

# Monitoring of activity-driven trafficking of endogenous synaptic proteins through proximity labeling

**Carlos Pascual-Caro, Jaime de Juan-Sanz**[ID]*

Paris Brain Institute (ICM). Sorbonne University, Inserm, CNRS, APHP, Hôpital de la Pitié Salpêtrière, Paris, France

* jaime.dejuansanz@icm-institute.org

**Data Availability Statement:** All relevant data are within the paper and its Supporting Information files.

## Abstract

To enable transmission of information in the brain, synaptic vesicles fuse to presynaptic membranes, liberating their content and exposing transiently a myriad of vesicular transmembrane proteins. However, versatile methods for quantifying the synaptic translocation of endogenous proteins during neuronal activity remain unavailable, as the fast dynamics of synaptic vesicle cycling difficult specific isolation of trafficking proteins during such a transient surface exposure. Here, we developed a novel approach using synaptic cleft proximity labeling to capture and quantify activity-driven trafficking of endogenous synaptic proteins at the synapse. We show that accelerating cleft biotinylation times to match the fast dynamics of vesicle exocytosis allows capturing endogenous proteins transiently exposed at the synaptic surface during neural activity, enabling for the first time the study of the translocation of nearly every endogenous synaptic protein. As proof-of-concept, we further applied this technology to obtain direct evidence of the surface translocation of noncanonical trafficking proteins, such as ATG9A and NPTX1, which had been proposed to traffic during activity but for which direct proof had not yet been shown. The technological advancement presented here will facilitate future studies dissecting the molecular identity of proteins exocytosed at the synapse during activity, helping to define the molecular machinery that sustains neurotransmission in the mammalian brain.

## Introduction

Transmission of information in the brain relies on a series of orchestrated trafficking events at presynapses that allow synaptic vesicles (SVs) to fuse to the plasma membrane and release neurotransmitters [1–3]. Such events transiently remodel the molecular composition of the presynaptic surface, briefly exposing hundreds of proteins that would otherwise be present in low abundance at the synaptic plasma membrane. The proteins that are translocated must be retrieved back after activity through endocytosis in a process that places a significant energy burden on firing synapses [4–7]. Given that optimizing energy usage has played a major role in the evolution of nervous systems [8,9], it is widely assumed that only proteins whose

**Funding:** This work was made possible by the Paris Brain Institute Diane Barriere Chair in Synaptic Bioenergetics awarded to Jaime de Juan-Sanz. Our funding sources are an ERC Starting Grant SynaptoEnergy (European Research Council, ERC-StG-852873), 2019 ATIP-Avenir Grant (CNRS, Inserm) and a Big Brain Theory Grant (ICM) awarded to J.d.J-S. C.P-C. is the recipient of a postdoctoral fellowship from the Fundacion Martin Escudero. J.d.J-S is a permanent CNRS researcher and a FENS-Kavli Scholar. The funders had no role in study design, data collection and analysis, decision to publish, or preparation of the manuscript.

**Competing interests:** The authors have declared that no competing interests exist.

**Abbreviations:** ATG9A, Autophagy Related 9A protein; BSA, bovine serum albumin; FDR, false discovery rate; NPTX1, Neuronal Pentraxin 1; PMSF, phenylmethylsulfonyl fluoride; ROI, region of interest; RT, room temperature; SV, synaptic vesicle; WGA, wheat germ agglutinin.

function is essential to sustain neurotransmission are trafficked to the synaptic cleft during firing, making the study of translocating synaptic proteins of particular interest in the understanding of information processing in the brain.

To visualize protein translocation to the presynaptic surface, the field has classically relied on tagging proteins of interest with pHluorin, a pH-sensitive variant of GFP that enables robust quantification of exocytosis events [10–16]. The fluorescence of pHluorin is quenched at rest by the acidic pH found in the lumen of synaptic vesicles. During neuronal activity, synaptic vesicles fuse to the presynaptic surface and expose pHluorin to the extracellular pH, increasing its fluorescence approximately 100 times [10]. However, this strategy requires overexpressing constructs of interest, which could alter the localization or biological properties of the protein being studied [17]. Alternatively, activity-driven translocation of endogenous proteins can be quantified using specific antibodies that recognize luminal domains of translocated proteins, which only become accessible in live cells if the protein is translocated to the surface [18,19]. However, such antibodies are rare and only available for a few well-studied proteins, restricting the generalizability of this approach. A possible alternative strategy to label any endogenous surface protein could be to use impermeant labeling reagents such as biotin sulfo-NHS (N-hydroxysulfosuccinimide), which biotinylate all surface proteins in a cell [20–23]. However, the low labeling efficiency of this method requires tens of minutes of labeling reaction, making it inadequate to capture proteins that are exo- and endocytosed in just a few seconds. Moreover, this labeling approach would encompass the entire neuronal surface, making it suboptimal for capturing changes in protein abundance occurring exclusively within synaptic clefts.

Being an unbound neuronal compartment, the isolation of the synaptic cleft by classic biochemical fractionation has remained challenging for decades. However, recent work solved this problem by leveraging the use of proximity-labeling technologies specifically targeted to this locale [24]. Proximity labeling leverages the activity of genetically encoded biotinylating enzymes, such as APEX or HRP, which are fused to proteins of interest to label selectively surrounding proteins at subcellular sites of interest [24–28]. The APEX or HRP biotinylating activity remains silent until these enzymes are in the presence of $H_2O_2$ and biotin phenol for 1 min, which generates biotin-phenoxyl radicals whose very short half-life is approximately 1 millisecond, restricting their labeling radius to 50 to 60 nanometers. By expressing HRP at the synaptic surface fused to postsynaptic proteins of interest and restricting biotinylating activity to surface-exposed proteins using a cell-impermeant biotin phenol, this technology now allows specific isolation of synaptic cleft proteins at rest [24].

Here, we developed a novel strategy to leverage the use of synaptic cleft proximity biotinylation for capturing the translocation of any possible endogenous synaptic protein during neuronal activity. By reducing biotinylating times to just 15 s, we show the quantitative change in abundance of known endogenous synaptic vesicle proteins at the synaptic cleft during firing, showing that activity induces their exocytosis at synapses. We leveraged the use of this technology to explore the trafficking of endogenous noncanonical proteins at the presynapse, providing for the first time a direct measurement of activity-driven trafficking of endogenous Autophagy Related 9A protein (ATG9A) and Neuronal Pentraxin 1 (NPTX1). Using novel pHluorin-tagged constructs for these proteins, we confirmed their translocation during activity within the presynapse and demonstrated that proximity labeling readouts showed comparable quantitative changes in surface protein abundance when compared to pHluorin results. The development of our novel approach will enable dissecting the molecular identity of trafficked proteins at the synapse and provides a robust technical framework for future studies deciphering the role of protein trafficking in sustaining synaptic function in health and disease.

## Results

### Fast synaptic cleft proximity labeling enables detecting activity-driven exocytosis of endogenous proteins

Previous work using proximity-labeling identified hundreds of proteins belonging to the synaptic cleft [24]. However, when we analyzed these published data sets using ontology analysis of the synapse [29], we found that synaptic vesicle proteins could not be identified as part of the cleft proteome (S1A Fig) despite localizing at the presynaptic surface of firing neurons [2,30]. We thus reasoned that synaptic cleft isolation using proximity labeling in primary neurons provides a relatively silent state of the synapse in which most synaptic vesicle proteins are located intracellularly. We hypothesized that experimentally increasing neuronal activity should allow increasing the surface abundance of endogenous proteins that undergo exocytosis, allowing us to detect them at the synaptic surface during firing. Dissociated neurons in culture form functional synapses [31–33] and favor rapid delivery and washout of biotin-phenol, $H_2O_2$ and subsequent peroxidase quenchers required for proximity labeling [24,34]. We reasoned that incorporating high potassium chloride (KCl) in the biotinylation solution should simultaneously depolarize the neurons and induce labeling of newly exposed proteins, optimizing the coordination of both processes (Fig 1A).

While an extensive body of work using proximity labeling has defined that the optimal biotinylating time for APEX or HRP is approximately 60 s [24–27], the translocation of the entire SV pool is expected to occur much faster when neurons are depolarized [35]. For high signal-to-noise detection of SV translocation, we reasoned that biotinylating times should be adapted to the time it takes to mobilize the SV pool during depolarization, as otherwise increasing labeling times would only result in unspecific background labeling because all SV proteins of interest are already translocated and labeled. We first quantified the dynamics of SV translocation by KCl using synaptophysin-pHluorin (Syp-pH), aiming to determine the duration required for SV translocation to reach a stable plateau to dictate optimal times for subsequent biotinylation experiments. We applied KCl during 15, 30, or 60 s and tracked the kinetics of SV exocytosis. We found that maximal mobilization of the SV pool occurred before 15 s in the presence of KCl in our system (Fig 1B), as previously reported [35]. Measuring the fraction of time in which SVs are still being translocated in the presence of KCl indicated that shorter times should favor specific biotinylation of translocated proteins (Figs 1C and S1B).

We next located HRP in the synaptic cleft using lentiviruses that express both LRRTM1-HRP and LRRTM2-HRP [24]. First, we confirmed that expressing both LRRTM-HRP constructs did not alter trafficking properties of synaptic vesicles. We combined Syp-pH with field stimulation and quantified activity-driven exocytosis in both control and transduced neurons, finding no difference in SV exocytosis (Fig 1D) or SV pool size (S1C Fig). When exocytosis was driven by a pulse of 15 s exposure to KCl, we also found no significant differences in SV translocation (Fig 1E), confirming that lentiviral transduction of LRRTM proteins does not alter SV trafficking. Next, we corroborated that the biotinylation reaction also worked robustly in the presence of KCl using western blot and streptavidin-HRP. This approach revealed all biotinylated proteins, which appeared as a smeared band spanning most molecular weights as expected [24,26,27] (Fig 1F). Quantification of biotinylation rates from total lysates revealed no significant differences (S2A and S2B Fig), similar to the quantification of biotinylation rates from streptavidin-based synaptic cleft enrichments (S2C and S2D Fig). These series of controls confirmed that this system should enable capturing differences in endogenous translocated proteins during activity.

To test our hypothesis, we next triggered simultaneously depolarization and biotinylation during 15 or 30 s and evaluated changes in cleft surface abundance of different proteins

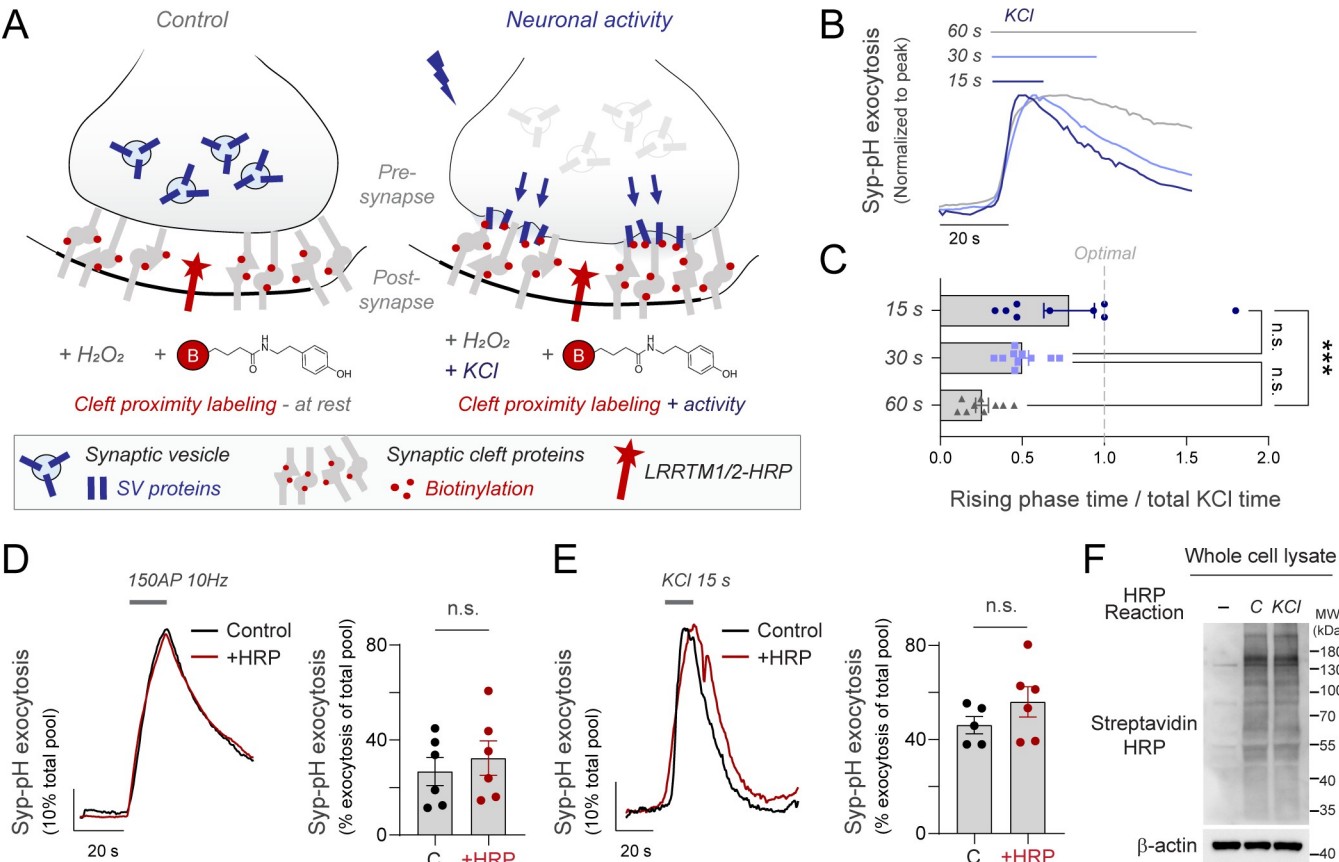

**Fig 1. Approach to visualize activity-driven trafficking of endogenous synaptic proteins through proximity labeling. (A)** Design of experimental strategy using synaptic cleft proximity labeling to isolate endogenous proteins translocated to the synaptic surface during neuronal activity. Biotinylation occurs at pre- and post-synaptic surfaces using an impermeant biotin-XX tyramide reagent (BxxP, shown in red). **(B)** Synaptic vesicle exocytosis using synaptophysin-pHluorin (Syp-pH) in different times of potassium chloride (KCl) treatment shows that the SV releasable pool is exhausted in less than 15 s. Average traces shown are normalized to the peak response. **(C)** Comparison of the fraction of time in the presence of KCl in which pHluorin signal is rising. A theoretical optimal time should translocate the entire SV pool in the same time KCl is applied (indicated by gray dashed line). **(D)** Example trace of synaptic vesicle cycling using Syp-pH during 150 action potentials (AP) fired at 10 Hz using field stimulation. Right panel shows exocytosis as the percentage of Syp-pH mobilized to the surface from the total pool. **(E)** Example trace of synaptic vesicle cycling using Syp-pH during 15 s of KCl exposure. Right panel shows exocytosis as the percentage of Syp-pH mobilized to the surface from the total pool. **(F)** Representative image of biotinylation rates obtained with streptavidin-HRP (Streptav-HRP) in neurons not expressing LRRTM-HRP (-), or neurons expressing LRRTM-HRP in control (C) or depolarization (KCl) conditions. β-actin is shown as a loading control. S1 Table shows the number of experiments and replicates, means and error, and statistical tests used.

known to undergo activity-driven translocation at synapses, such as vGlut1, Synaptophysin 1, or Synaptotagmin 1. These experiments revealed significant increases in these proteins at the synaptic cleft if the reaction occurred during 15 s (Fig 2A and 2B). However, we could not reliably capture their translocation if the reaction lasted for 30 s (Fig 2A and 2B). Using 30 s of biotinylation typically reported more intense bands for synaptic proteins at rest, even for those expected to have very low surface fraction, like vGlut1 [36]. This is likely the consequence of increased biotinylation times, together with the fact that streptavidin enrichment is performed from hundreds of micrograms of protein (see Materials and methods), revealing even small amounts of protein when combining sensitive antibodies and western blot. These results indicate that coordinating short labeling times and translocation dynamics is essential to detect relative changes in surface abundance of trafficking proteins.

Experiments showed higher variability when activating neurons for 15 s, perhaps due to the experimental variability in exchanging buffers for triggering and stopping the biotinylation

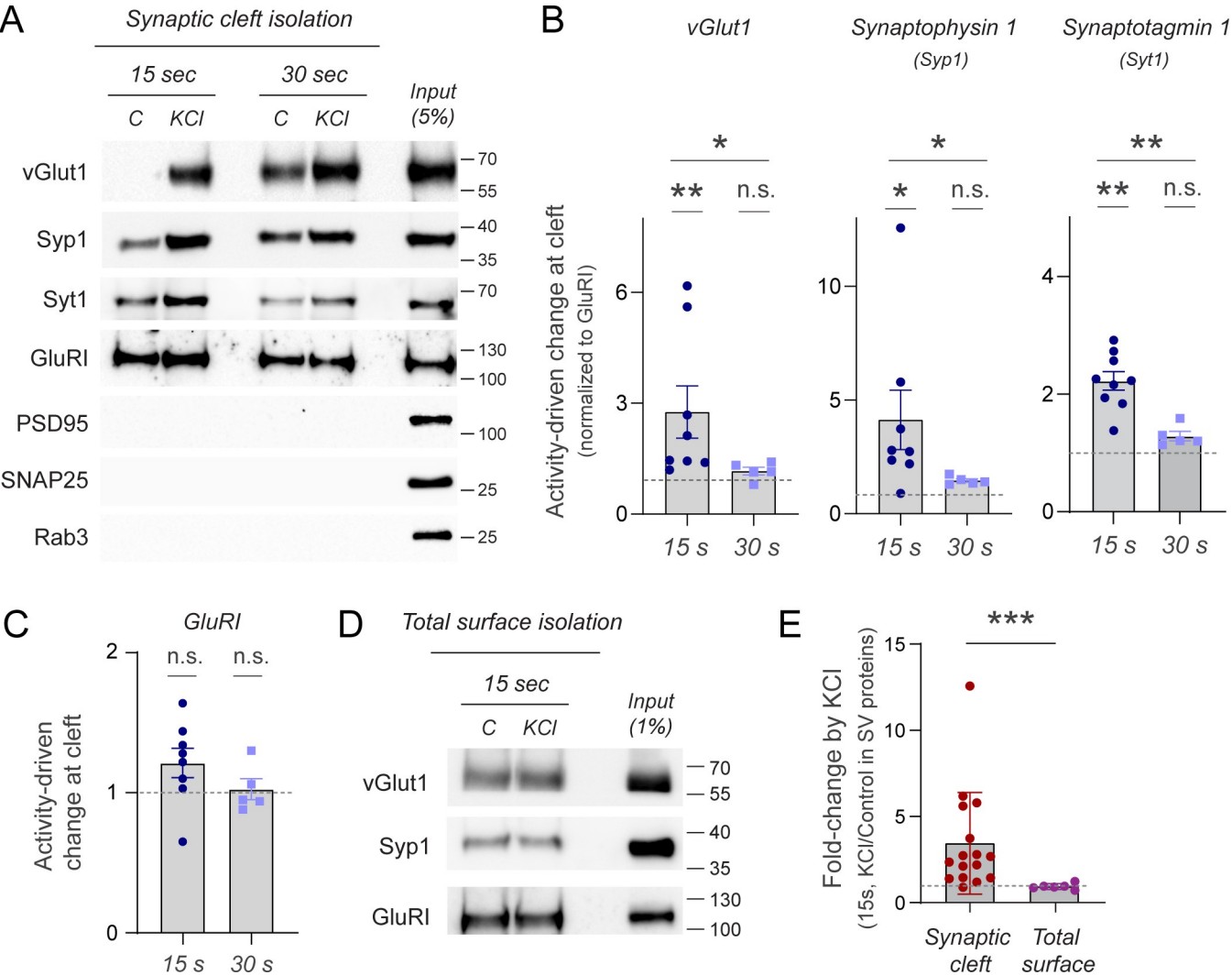

**Fig 2. Activity-driven trafficking of endogenous synaptic proteins through synaptic cleft proximity labeling. (A)** Representative western blot of the isolation of synaptic cleft proteins using LRRTM-HRP proximity biotinylation during 15 or 30 s in the presence or absence of KCl. Subsequent streptavidin-based purification enabled western blotting against vesicular glutamate transporter 1 (vGlut1), synaptophysin 1 (Syp1), synaptotagmin 1 (Syt1), Glutamate Receptor 1 (GluRI), postsynaptic density protein 95 (PSD95), Synaptosomal-Associated Protein, 25 kDa (SNAP25), and Ras-related protein Rab-3A (Rab3), showing changes in synaptic surface abundance only in surface-exposed SV proteins; 100 μg protein were used for the Streptavidin enrichment beads of this approach. **(B)** Quantification of relative changes in synaptic surface abundance in vGlut1, Syp1, and Syt1 shows that 15 s of reaction allows detecting a significant surface increase of both proteins by depolarization. Signals are normalized to GluRI immunoreactivity. Dashed line indicates no change (KCl/Control = 1). **(C)** Quantification of relative changes in synaptic surface abundance of GluRI during 15 or 30 s shows no change in GluRI at the synaptic surface during depolarization. **(D)** Representative western blot of the isolation of neuronal surface proteins by proximity biotinylation using a plasma membrane-localized HRP (HRP-TM) during 15 s in the presence or absence of KCl, 500 μg protein were used for the Streptavidin enrichment beads of this approach. Note that no apparent changes are observed in either vGlut1 or Syp1, in contrast to synaptic cleft isolation and that detection by HRP-TM is favored by loading samples obtained from 100 times more protein than the loaded input. **(E)** Quantification of relative changes in surface abundance of synaptic vesicle proteins (vGlut and Syp1) at the synaptic cleft or at the global neuronal surface during 15 s of proximity labeling reaction. Increases in SV protein surface abundance are significantly detected when proteins are isolated from the synaptic cleft. Signals are normalized to GluRI immunoreactivity. Dashed line indicates no change (KCl/Control = 1). S1 Table shows the number of experiments and replicates, means and error, and statistical tests used.

reaction, which should have a larger impact when using shorter reactions. However, despite the variability observed, this approach consistently reported activity-driven translocation changes of endogenous vGlut1, Synaptophysin 1, and Synaptotagmin 1. To control for biotinylation rates in each experimental condition, we looked for proteins known to be localized at the synaptic cleft but whose abundance remains unchanged during the time used for KCl

exposure. We reasoned that such a control should provide a quantitative readout of the success of the reaction in each condition tested. We quantified the synaptic abundance of postsynaptic Ionotropic Glutamate Receptor 1 (GluRI) in response to KCl during either 15 or 30 s of depolarization (Fig 2C) and found no significant change, providing an ideal control for biotinylation rates at the cleft. Of note, isolated biotinylated proteins were subjected to protocols that disassemble detergent-insoluble synaptic structures to remove from the sample unspecific cytosolic proteins that could be potentially isolated through protein–protein interactions and not direct biotinylation (see Materials and methods and S1 Protocol). We confirmed that this is the case by observing no synaptic cleft isolation of PSD95, a major contaminant that becomes isolated if postsynaptic densities are not properly disrupted [24] (Fig 2A).

To also confirm that increases observed did not arise from artifactual intracellular labeling induced by KCl, we next examined changes in surface abundance of SNAP25 and Rab3, two cytosolic SV proteins that are never exposed to the synaptic surface. Western blot analysis against SNAP25 or Rab3 demonstrated that, while these proteins are clearly expressed and found in the sample before streptavidin-based isolation (see input, Fig 2A), they are not biotinylated by surface-localized HRP, as no bands can be detected in either control or KCl conditions. Moreover, using immunofluorescence, we confirmed that no biotinylation occurred intracellularly under our labeling conditions. We expressed APEX2 in the cytosol, performed the biotinylating reaction using BxxP as a substrate, permeabilized the cells and stained them with streptavidin conjugated to Alexa Fluor 647, which stains specifically biotinylated proteins. When expressing cytosolic APEX2, we observed negligible labeling compared to the same reaction with surface-expressed LRRTM-HRP, supporting the conclusion that BxxP cannot act as a biotin substrate intracellularly and that no significant biotinylation occurs inside neurons (S2E Fig).

Lastly, we confirmed that the synaptic cleft localization of HRP was essential to capture translocation of synaptic proteins. We expressed HRP in the entire surface of the neuron using HRP-TM [24] and quantified activity-driven changes in surface abundance of synaptic vesicle proteins during 15 s. These experiments revealed that only using cleft-localized proximity labeling enabled capturing translocation of endogenous SV proteins reliably (Fig 2D and 2E). Of note, in these experiments we had to use larger amounts of input sample to be able to enrich synaptic proteins sufficiently for western blot detection (see Materials and methods), which further shows the inadequacy of expressing HRP in the entire surface of the neuron if one wants to detect synaptic proteins. Taken together, these results demonstrate that by carefully coordinating biotinylation and depolarization at synaptic clefts, and reducing labeling times to limit nonspecific labeling, proximity labeling enables the specific isolation and quantification of endogenous translocated proteins to the synaptic surface during neuronal activity.

## Proximity labeling confirms activity-driven translocation of endogenous proteins previously predicted to traffic during firing

By providing a flexible strategy to study synaptic translocation of proteins, our new experimental approach should facilitate directly assessing the degree to which certain proteins, hypothesized to participate in activity-driven translocation at synapses, indeed engage in such a process. As a proof-of-concept, we sought to test some of those predictions we identified in the literature.

Neuronal pentraxins or NPTXs are presynaptic organizers located in the excitatory synaptic cleft involved in synapse formation [37] and maintenance [38], synaptic function [39,40], and activity-driven plasticity [41–43]. The surface abundance of one of the members of the family, NPTX1, governs AMPA receptor clustering and neurotransmission strength [37,44] and its

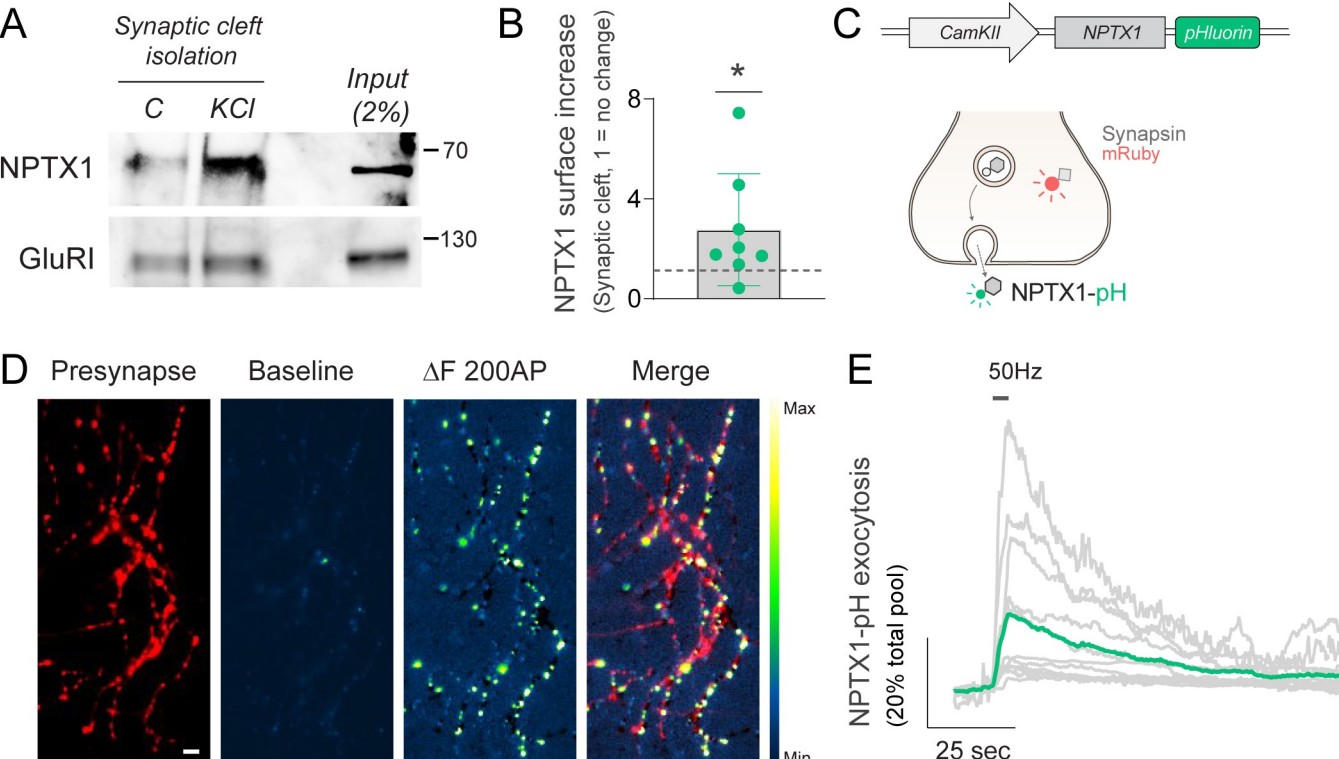

**Fig 3. Activity drives presynaptic surface exposure of Neuronal Pentraxin 1. (A)** Representative western blot of the isolation of synaptic cleft during 15 s in the presence or absence of KCl. Subsequent streptavidin-based purification and western blot show increases in synaptic surface abundance of NPTX1 with no change in GluRI, 250 µg protein were used for the Streptavidin enrichment beads of this approach. **(B)** Quantification of relative changes in synaptic surface abundance in NPTX1 shows a significant increase by KCl. Signals are normalized to GluRI immunoreactivity. Dashed line indicates no change (KCl/Control = 1). **(C)** Design to express NPTX1 fused to pHluorin. Schema shows expected changes in fluorescence of NPTX1-pH during translocation and co-expression with Synapsin-mRuby to identify presynaptic sites. **(D)** Representative image of a presynaptic arborization of a neuron expressing Synapsin-mRuby (presynapse) and NPTX1-pH. Baseline NPTX1-pH fluorescence is relatively low, but becomes significantly fluorescent at synaptic sites after field stimulation of 200 AP at 50 Hz. Image labeled as ΔF 200 AP shows the change in fluorescence driven by activity. Merge shows ΔF 200 AP superposed to presynaptic labeling by Synapsin-mRuby. Scale bar = 4.8 µm. Pseudocolor bar indicates intensity range in NPTX1-pH images. **(E)** Quantification of NPTX1-pH exocytosis in individual neurons in response to electrical stimulation, shown in gray. Green trace shows the average response. Traces are normalized to the total NPTX1-pH mobilizable pool, obtained by application of $NH_4Cl$ (pH 7.4). S1 Table shows the number of experiments and replicates, means and error, and statistical tests used.

dysfunction is associated with several neurological diseases [45]. NPTX1 is released from axons by increases in astrocyte-secreted glypican-4 (GPC4) [37] and it has been hypothesized that its translocation could be also increased by neuronal activity [45]. However, no evidence exists, to our knowledge, testing this hypothesis. We thus examined whether we could detect activity-driven synaptic translocation of endogenous NPTX1 using cleft proximity labeling. Using fast HRP biotinylation at the synaptic cleft, we found robust activity-driven transloca-tion of endogenous NPTX1 (Fig 3A). These experiments showed a ~2.5-fold increase in the number of molecules found at the synaptic cleft during just 15 s of firing (Fig 3B), indicating that endogenous NPTX1 is efficiently exposed to the synaptic surface during activity. To con-firm this result in single synapses, we fused pHluorin to the c-terminal end of NPTX1 (NPTX1-pH) and selectively expressed this construct in excitatory neurons using the CamKII promoter [46] (Fig 3C). Co-expression of NPTX1-pH together with the presynaptic marker Synapsin-mRuby confirmed the expected localization of NPTX1-pH at presynapses of live neurons, which we visualized by applying $NH_4Cl$ to de-acidify internal compartments contain-ing NPTX1 and reveal the total amount of the protein (S3A Fig). These experiments showed

that approximately 90% of NPTX1-pHluorin was on average located intracellularly (S3B Fig), supporting the idea that activity can significantly drive its translocation to the surface on demand. Using electrical field stimulation, we induced the firing of 200 action potentials at 50 Hz and observed a robust translocation of NPTX1-pH at synapses (Fig 3D and 3E). We chose to study NPTX1-pH translocation using field stimulation as it enhances reproducibility of the stimulus compared to KCl application. However, a 15 s application of KCl also drove strong translocation of NPTX1-pH to the synaptic surface, as expected (S3C Fig). These results confirmed that NPTX1 is translocated to the synaptic cleft during neuronal activity and further supported the validity of our results showing endogenous NPTX1 translocation using proximity labeling. Moreover, these results also support the growing evidence that certain secreted trans-synaptic organizers, including LGI1 [16] and Cerebellin 1 [12], can be translocated to the synaptic surface during neuronal activity.

We sought to further test our system with additional proof-of-concept experiments. In synapses, macroautophagy/autophagy responds to increased neuronal activity states [47–50]. Previous work proposed that ATG9A, the only transmembrane protein in the core autophagy pathway, undergoes exo-endocytosis in an activity-dependent manner in presynapses to coordinate the interplay between activity and autophagy [51,52]. However, a direct visualization of endogenous ATG9A translocation to the synaptic surface during activity has not yet been shown, to our knowledge. We used fast proximity labeling at the synaptic cleft and found robust activity-driven translocation of endogenous ATG9A, which increased approximately 1.5-fold the number of molecules found at the synaptic cleft in just 15 s (Fig 4A and 4B). To confirm this result using imaging, we expressed ATG9A-pHluorin [53] in neurons and examined its capacity for translocation in presynapses during field stimulation (Fig 4C). Similar to NPTX1, a large fraction of ATG9A-pHluorin was located intracellularly at presynapses (S3D and S3E Fig), enabling the possibility of translocation on demand. Our experiments showed that ATG9A was robustly translocated to the synaptic surface in response to 200 AP fired at 50 Hz (Fig 4D and 4E) and during 15 s of KCl application (S3F Fig), validating previous predictions [51,52] and our results obtained using proximity labeling. ATG9A is known to localize at the presynapse [51,52] but in a distinct class of vesicles which are not bona fide SVs [54,55]. These results suggest that despite not being localized in SVs, ATG9A-vesicles robustly undergo activity-driven exocytosis.

We next contrasted activity-driven changes in surface protein levels obtained from imaging pHluorin-tagged constructs with those obtained through proximity labeling (Fig 4F). This analysis revealed that proteins exhibiting more pronounced translocation changes through proximity labeling, such as Synaptophysin 1 (Fig 4G), also demonstrated greater changes when assessed using pHluorin imaging during electrical stimulation (Fig 4H) or KCl exposure (S3G Fig). These results demonstrate that fast synaptic cleft proximity biotinylation is ideally positioned to study activity-driven translocation of endogenous synaptic proteins.

## Discussion

Experimentally demonstrating that endogenous proteins of interest are translocated to the synaptic surface during the fast timescales of neuronal activity has remained a challenge in the field. Here, we optimized and implemented the use of proximity labeling technologies to capture fast trafficking events of endogenous proteins at the synaptic cleft of firing neurons. Exocytosis of synaptic vesicles occurs in time scales of seconds, and it is subsequently followed by fast retrieval of exposed proteins back to the intracellular space [2,5,6,30]. To successfully label proteins undergoing such fast translocation, we reasoned that only biotinylating enzymes with fast kinetics could be used. Biotinylating ligases that are compatible with labeling in vivo, such

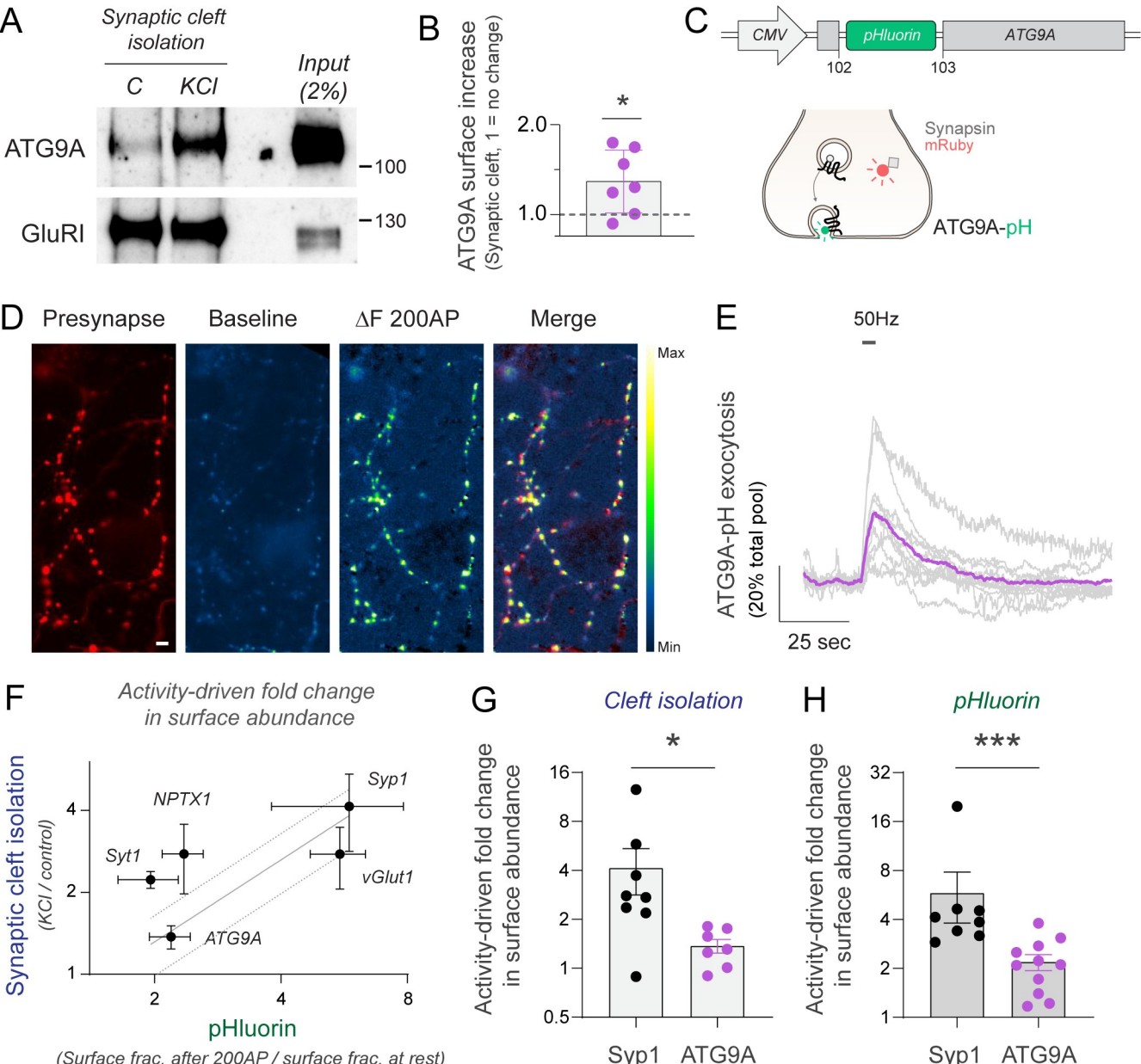

**Fig 4. Activity drives presynaptic surface exposure of ATG9A. (A)** Representative western blot of the isolation of synaptic cleft during 15 s in the presence or absence of KCl. Subsequent streptavidin-based purification and western blot show increases in synaptic surface abundance of ATG9A with no change in GluRI; 250 μg protein were used for the Streptavidin enrichment beads of this approach. **(B)** Quantification of relative changes in synaptic surface abundance in ATG9A shows a significant increase by KCl. Signals are normalized to GluRI immunoreactivity. Dashed line indicates no change (KCl/Control = 1). **(C)** Design to express ATG9A fused to pHluorin by Campisi and colleagues [53]. Schema shows expected changes in fluorescence of ATG9A-pH during translocation and co-expression with Synapsin-mRuby to identify presynaptic sites. **(D)** Representative image of a presynaptic arborization of a neuron expressing Synapsin-mRuby (presynapse) and ATG9A-pH. Baseline ATG9A-pH fluorescence is relatively low, but becomes significantly fluorescent at synaptic sites after field stimulation of 200 AP at 50 Hz. Image labeled as ΔF 200 AP shows the change in fluorescence driven by activity. Merge shows ΔF 200 AP superposed to presynaptic labeling by Synapsin-mRuby, showing significant overlap. Scale bar = 4.8 μm. Pseudocolor bar indicates intensity range in ATG9A-pH images. **(E)** Quantification of ATG9A-pH exocytosis in individual neurons in response to electrical stimulation, shown in gray. Purple trace shows the average response. Traces are normalized to the total ATG9A-pH mobilizable pool, obtained by application of $NH_4Cl$ (pH 7.4). **(F)** Quantification of surface abundance changes in ATG9A, NPTX1, vGlut1, Syt1, and Syp1 by synaptic cleft isolation and pHluorin imaging. Changes in pHluorin abundance are quantified by measuring relative changes in surface fraction induced by 200 AP at 50 Hz, while changes in the synaptic cleft of endogenous proteins are measured as in (B). A simple linear regression is shown (continuous gray line) along with its 90% confidence bands (dotted lines). **(G)** Quantification of relative changes in synaptic surface abundance by synaptic cleft isolation in Syp1 and ATG9A shows quantitatively more translocation of Syp1 by activity. Note that changes in Syp1 are already shown in Fig 2B and ATG9A changes are shown in (B), but are plotted here to ease visualization of comparison. **(H)** Quantification of relative changes in synaptic surface abundance by imaging of pHluorin-tagged Syp1 and ATG9A, which shows quantitatively more translocation of Syp1 by activity similar to data from (G). S1 Table shows the number of experiments and replicates, means and error, and statistical tests used.

as BioID [56] or TurboID [57], require at least 10 min for successful labeling, losing the temporal resolution required for labeling fast exocytosis events. Alternatively, APEX2 and HRP are much faster and have been used for as short as 1 min [24–27,58]. However, while 1 min is 10 times faster than typical times used for BioID or TurboID, it is still suboptimal for capturing fast exocytic events that occur in seconds. When comparing APEX to HRP, the latter presented faster kinetics and greater biotinylating capabilities [59]. We leveraged the optimal biotinylating capacity of HRP to detect fast exocytosis events and demonstrated that HRP can potently label surrounding proteins even if the reaction is shortened 4-fold. These results open the possibility of using proximity labeling for the first time to capture fast translocation events specifically at the synapse and indicate that HRP will be an ideal labeling enzyme to detect fast trafficking events of endogenous proteins in other fields of cellular biology.

Alternative strategies for detecting trafficking of endogenous proteins in live neurons, such antibody labeling of SV luminal protein domains during translocation, can provide high sensitivity of detection if antibodies are robust and specific [18,19]. However, relying solely on antibodies is not without limitations: (1) epitopes must be in parts of the protein exposed to the surface; and (2) antibodies used must have a strong binding affinity and specificity for efficient labeling in just a few seconds. Proximity labeling offers an alternative to both these problems, as biotinylation (a) can occur in any surface-exposed part of the protein; and (b) western blot allows quantifying specific bands corresponding to the protein of interest, even if the antibody recognizes other targets and shows additional bands. Moreover, the affinity of the antibody for its target does not need to be exceptional for successful western blotting, while it must be highly potent for live cell labeling in a few seconds. Lastly, western blot uses significantly less antibodies, making proximity labeling much less costly and easier to implement.

Different technologies already exist to isolate surface proteins in neurons and other cell types, including biochemical isolation of plasma membranes [60] or global biotinylation strategies in which all surface proteins in all locations are labeled for subsequent isolation [61]. However, these approaches can only report relatively slow changes occurring during minutes or hours. Alternatively, surface-localized HRP could facilitate faster labeling at the entire surface of cells given its high biotinylating capacity [59]. HRP can be localized to the cell surface by either applying purified HRP fused to the glycan-binding domain of wheat germ agglutinin (WGA-HRP) [62], which makes it bind to the plasma membrane, or by expressing a construct that localizes HRP to the surface (HRP-TM) [24]. However, we tested the ability of the latter to report trafficking changes in neurons depolarized for 15 s and did not find robust changes in surface localization of known trafficking synaptic proteins. These results suggest that global biotinylation strategies, even if optimized for fast labeling kinetics, are still suboptimal in detecting activity-driven exocytosis events occurring specifically at the synapse. Thus, both fast labeling and specific synaptic localization of HRP are required to successfully capture activity-driven translocation events of synaptic proteins. These results suggest that future strategies using proximity labeling to capture translocation of endogenous proteins in other fields of cellular biology will present improved signal-to-noise if biotinylating proteins are targeted to sites of translocation.

Our novel strategy presented here enables direct measurements of the translocation of any endogenous protein of interest at the synapse. This includes presynaptic proteins, as shown here, but should also enable detecting trafficking events occurring at the postsynapse during 15 s of depolarization. However, our experiments quantifying total changes in synaptic cleft biotinylation with or without activity did not reveal significant differences, suggesting that the total amount of proteins exposed at the cleft after SV exocytosis may not be significant enough to increase the already dense proteome of the synaptic cleft, thus contributing only a relatively small percentage change that may be beyond detection using western blot. In addition, we also

optimized our approach for detecting fast events occurring at synapses, but it is worth noticing that perhaps slower synaptic trafficking pathways may not be well represented in our samples and their detection could benefit from additional optimization by coordinating labeling times with their trafficking properties. Lastly, using KCl for stimulation does not optimally recapitulate neuronal firing and thus future developments in using more physiological stimuli for the neuronal activation of millions of cultured neurons, such as optogenetics, may improve the quality of the results.

Interestingly, our method should enable distinguishing functionally, rather than biochemically [63,64], whether a protein is located in a trafficking organelle. Increasing evidence shows that noncanonical synaptic vesicle proteins can translocate to the synaptic surface during neuronal activity, yet direct proof of such trafficking occurring for the endogenous proteins remains unproven [12–14,16]. As a proof of concept, we show that endogenous NPTX1 and ATG9A are translocated to the synaptic surface, providing important biological insight into the functioning of these proteins at the synapses. We confirmed these experiments using pHluorin tagged versions of both proteins, indicating that results on activity-driven translocation of endogenous synaptic proteins using proximity labeling are reproducible using established techniques and show quantitatively similar results. Synaptic cleft proximity labeling offers a reproducible, cost-effective, and readily deployable method for directly observing the endogenous translocation of synaptic proteins of interest, enabling the field to investigate the translocation of canonical and noncanonical synaptic proteins. Moreover, by demonstrating the robust labeling capability of HRP for surrounding proteins within a few seconds, our study establishes a crucial experimental framework for implementing strategies to quantify rapid translocation of endogenous proteins across various biomedical domains beyond cellular neuroscience. Future work on developing and scaling up this method for resolving the activity-driven synaptic cleft proteome will be of great interest and may provide an unbiased view of the remodeling of the synaptic cleft during activity, perhaps identifying novel translocating proteins whose function may contribute to synaptic physiology and neurotransmission.

## Materials and methods

### Resources table

Shown in Table 1.

### Animals

The wild-type rats used in this study were of the Sprague Dawley strain Crl:CD(SD), bred globally by Charles River Laboratories in accordance with the International Genetic Standard Protocol (IGS). The execution of all experimental procedures strictly adhered to the guidelines set forth by the European Directive 2010/63/EU and the French Decree No. 2013–118, which concern the protection of animals used in scientific research. Approval for the experiments was obtained from the local ethics committee, Comité d'éthique N˚5 Darwin (Approval protocol number P186R).

### Primary co-culture of postnatal cortical neurons and astrocytes for imaging experiments

Imaging experiments were performed in primary co-cultures of cortical neurons and astrocytes. P0 to P2 rats of mixed gender were sacrificed and their brains were dissected in a cold HBSS-FBS solution (1X HBSS + 20% fetal bovine serum) to isolate the cerebral cortexes, which were subsequently cut into small pieces for further digestion and dissociation. This initial step

**Table 1. Resources table.**

| Reagent or resource | Source | Identifier |
|---|---|---|
| **Experimental models: Organisms/strains** | | |
| Sprague Dawley rat | Charles River | Strain: Crl:CD(SD) |
| **Antibodies** | | |
| Rabbit anti-beta Actin | Thermo Fisher Scientific | PA5-85271 |
| Mouse anti-GluA1 | UC Davis/NIH NeuroMab | 75327 |
| Rabbit anti-ATG9A | Abcam | 108338 |
| Rabbit anti-NPTX1 | Proteintech | 20656-1-AP |
| Mouse anti-SNAP25 | Synaptic Systems | 111111 |
| Mouse anti-Synaptophysin 1 | Synaptic Systems | 105011 |
| Mouse anti-VGLUT 1 | Synaptic Systems | 135011 |
| Rabbit anti-Synaptotagmin 1 | Synaptic Systems | 105103 |
| Mouse anti-Rab3 | Synaptic Systems | 107011 |
| Rabbit anti-PSD95 | Proteintech | 20665-1-AP |
| Rabbit anti-GFP | Thermo Fisher Scientific | A6455 |
| Mouse anti-V5 | Thermo Fisher Scientific | R960-25 |
| Goat anti-Rabbit IgG (H+L)-HRP | BioRad | 1706515 |
| Goat anti-Mouse IgG (H+L)-HRP | BioRad | 1706516 |
| Goat anti-Mouse IgG (H+L)-Alexa-Fluor488 | Thermo Fisher Scientific | A11001 |
| Goat anti-Rabit IgG (H+L)-Alexa-Fluor488 | Thermo Fisher Scientific | A11034 |
| **Recombinant DNA** | | |
| CamKII-NPTX1-pHluorin | This paper | Addgene Plasmid #217974 |
| CMV-mRuby3-Synapsin1a | Gift from Timothy A. Ryan | Addgene Plasmid #187896 |
| CMV-ATG9-pHluorin | Campisi and colleagues [53] | N/A |
| CMV-Synaptophysin-pHluorin | Pulido and colleagues [65] | N/A |
| CamKII-vGlut1-pHluorin | Bae and colleagues [46] | N/A |
| FSW-HRP-V5-LRRTM1 | Loh and colleagues [24] | Addgene Plasmid #82536 |
| FSW-HRP-V5-LRRTM2 | Loh and colleagues [24] | Addgene Plasmid #82537 |
| FSW-TM-HRP | Loh and colleagues [24] | Addgene Plasmid #82540 |
| CamKII-Synaptotagmin1-pHluorin | This paper | Addgene Plasmid #224030 |
| V5-GFP-APEX2-NES_pLX208 | Qin and colleagues [28] | Addgene Plasmid #202018 |
| **Chemicals, peptides, and recombinant proteins** | | |
| PierceStreptavidin Magnetic Beads | Thermo Fisher Scientific | 88817 |
| BxxP impermeant-biotin phenol | ApexBio Technology | A8012 |
| Pierce High Sensitivity Streptavidin-HRP | Thermo Fisher Scientific | 21130 |
| Streptavidin, Alexa Fluor 647 | Thermo Fisher Scientific | S21374 |
| Papain | Worthington Biochemical | LK003178 |
| N-21 | Bio-techne | AR008 |
| 6-cyano-7-nitroquinoxaline-2,3-dione (CNQX) | Tocris | 1045 |
| D,L-2-amino-5-phospho-novaleric acid (AP5) | Tocris | 0106 |
| Protease inhibitor cocktail | Merck | P8849 |
| Digitonin | Merck | D141 |
| **Commercial culture media** | | |
| Minimum Essential Medium | Thermo Fisher Scientific | 51200038 |
| Neurobasal Medium | Thermo Fisher Scientific | 21103049 |
| BrainPhys Neuronal Medium | STEMCELL Technologies | 05792 |
| Advanced DMEM | Thermo Fisher Scientific | 12634–010 |

*(Continued)*

**Table 1.** (Continued)

| Reagent or resource | Source | Identifier |
|---|---|---|
| **Critical commercial assays** | | |
| NEBuilder HiFi DNA Gibson Assembly Master Mix | New England Biolabs | E2621 |
| PureLink Expi Endotoxin-Free Maxi Plasmid Purification Kit | Thermo Fisher Scientific | A31231 |
| p24 ELISA ZeptoMetrix kit | Merck | 0801111 |
| Pierce Rapid Gold BCA Protein Assay Kit | Thermo Fisher Scientific | A53225 |
| Clarity ECL Western Blotting Substrate | BioRad | 1705060 |
| Clarity Max ECL Western Blotting Substrate | BioRad | 1705062 |
| **Software** | | |
| ImageJ | National Institute of Health | https://imagej.nih.gov/ij/ |
| GraphPad Prism 9.0.0 | GraphPad Software | https://www.graphpad.com/prism |
| Benchling | Benchling | https://benchling.com/ |
| Image Lab software 6.1.0 | BioRad | https://www.bio-rad.com/ |
| **Other** | | |
| SynGO ID convert tool | Koopmans and colleagues [29] | www.syngoportal.org |

was followed by 2 washes with 1X HBSS-FBS, 2 washes with 1X HBSS and an incubation in a trypsin-based digestion solution containing DNAse I (Merck, D5025) for 15 min at room temperature (RT). Next, Trypsin (Merck, T1005) was neutralized by the addition of HBSS-FBS solution, followed by 2 washes with 1X HBSS-FBS and 2 washes with 1X HBSS. Then, the tissue was transferred to a dissociation solution (1X HBSS, 5.85 mM $MgSO_4$) and was dissociated by gentle and continuous pipetting into single cells. Next, the cells were cleared by centrifugation at 13,000 rpm for 10 min at 4°C and the pellet was resuspended in 1X HBSS solution. The cells were then pelleted again by centrifugation at 13,000 rpm for 10 min at 4°C and resuspended in a homemade warmed plating media composed of MEM (Thermo Fisher Scientific, 51200038) supplemented with 20 mM Glucose (Merck, G8270), 0.1 mg/ml transferrin (Merck, 616420), 1% GlutaMAX (Thermo Fisher Scientific, 35050061), 24 μg/ml insulin (Merck, I6634), 10% FBS (Thermo Fisher Scientific, 10082147), and 2% N-21 (Bio-techne, AR008). Finally, cells were plated within sterilized cloning cylinders (Merck, C7983; 38,000 cells per cloning cylinder) attached to coverslips (Diameter = 25 mm, Warner instruments, 640705) previously coated with 0.1 mg/ml poly-ornithine (Merck, P3655). After 3 to 4 days, neuronal media was exchanged with a homemade feeding media, whose composition is as plating media but with 5% FBS and 2 μM cytosine β-d-arabinofuranoside (Merck, C6645) to limit glial growth. Primary co-cultures of cortical neurons and astrocytes were maintained at 37°C in a 95% air/5% $CO_2$ humidified incubator prior to the imaging experiments, which were carried out from DIV12 to DIV21.

### Primary culture of embryonic cortical neurons for proximity biotinylation

Proximity biotinylation experiments were performed in primary cultures containing exclusively neurons and no astrocytes. Pregnant wild-type Sprague Dawley Crl:CD (SD) rats were purchased from Charles River Laboratories. Euthanasia through $CO_2$ and embryonic sac extraction were performed by certified veterinarians at the Paris Brain Institute ICMice phenopark animal core facility on the 18th day of embryonic gestation. E18 embryos were sacrificed and their brains were dissected in a cold HBSS-FBS solution to isolate the cerebral cortexes. To obtain dissociated cortical neurons, the same protocol as for the postnatal culture was

followed, except for the digestion of the cortical tissue, which was carried out with a papain (Worthington Biochemical, LK003178) digestion solution containing DNAse I for 30 min at 37°C. After the final centrifugation, the pelleted cells were resuspended in the same growth medium used previously by Loh and colleagues [24], composed of MEM supplemented with 2 mM L-glutamine (Thermo Fisher Scientific, 25030024), Neurobasal medium (Thermo Fisher Scientific, 21103049), 5% FBS, 2% N-21, and 0.5% GlutaMAX. Lastly, 4 million neurons were plated per 10 cm dish, which were previously coated with 0.1 mg/ml poly-D-lysine (Merck, P2636). After 5 days in vitro, the culture medium was partially replaced with BrainPhys Neuronal Medium supplemented with 2% SM1 (STEMCELL Technologies, 05792), 12.5 mM Glucose, and 10 μM 5′-fluoro-2′-deoxyuridine (FUDR; Thermo Fisher Scientific, 10144760), an antimitotic drug that suppresses growth of glial cells. Subsequent to the initial replacement, the medium was refreshed at intervals of every 4 to 5 days.

## Gene constructs

NPTX1-pHluorin was designed to express pHluorin after the NPTX1 c-terminus using a linker previously shown to not affect the properties of LGI1, a different trans-synaptic organizer [16]. Using Gibson Assembly (NEBuilder HiFi DNA Assembly Master Mix, New England Biolabs, E2621), we used CamKII-LGI1-pH (Addgene Plasmid #185537) to replace LGI1 by rat NPTX1 (a gift from Dr. Jonathan Elegheert, Uniprot #P47971). Synaptotagmin1-pHluorin was synthesized using GeneArt Custom Gene Synthesis from Thermo Fisher Scientific and was subcloned under the CamKII promoter by replacing LGI1-pH[16]. A detailed map can be found in Addgene, plasmid number #224430. The plasmid to express mRuby3-Synapsin1a was a gift from Timothy A. Ryan (Weill Cornell Medical College, New York, Addgene #187896). ATG9-pH was a gift from Dr. Fabrice Morin (University of Rouen Normandy, France) and it is described in Campisi and colleagues [53]. Synaptophysin-pHluorin (Syp-pH) [65,66] was also a gift from Dr. Timothy A. Ryan. The construct expressing vGlut-pHluorin (vGlut1-pH) [46] under the CamKII promoter was a gift from Dr. Sung Hyun Kim (Kyung Hee University). The plasmid to express NES-APEX2-GFP was a gift from Alice Y. Ting (Stanford University, Addgene plasmid # 202018).

## Neuronal transfections with calcium phosphate method

Transfections were performed in 6- to 8-day-old cultures using the calcium phosphate method as previously described by Sankaranarayanan and colleagues [10] with slight modifications. Briefly, cultures were washed once with Advanced DMEM (Thermo Fisher Scientific, 12634–010), not supplemented, and incubated for 1 h at 37°C in the incubator. The mix transfection (DNA-$Ca^{2+}$-$PO_4$ precipitate) was prepared by mixing plasmid DNA (using 0.63 μg per dish of each DNA when co-transfecting 2 different plasmids), 2 M $CaCl_2$ (20×, for a final concentration of 100 mM) (Fisher, BP9742-10x5), UltraPure Distilled Water (Thermo Fisher, 10977–015), and 2X HEPES-buffered saline (2× composition is, in gr/L: 16 NaCl, 0.7 KCl, 0.38 $Na_2H$-$PO_4$x7 $H_2O$, 2.7 Glucose, 10 HEPES, pH = 7.20 adjusted at 37°C) by gently vortexing 8 times after each addition. Then, the mixture was allowed to sit in the dark for 20–30 min. Later, the precipitate was added onto the cells and incubated for 1 h at 37°C in the incubator. To stop the reaction, the cloning cylinders were removed, allowing the cells to be washed with surrounding feeding media. All experiments were performed between 4 and 13 days after transfection. This method leads to very sparse transfection in primary neuronal cultures, with around 1% efficiency that allows to study fluorescence changes in individual presynaptic arborizations of single neurons during field stimulation.

## Lentivirus production

The following genetic constructs for proximity labeling were a gift of Dr. Alice Ting (Stanford University): FSW-HRP-V5-LRRTM1 (Addgene #82536), FSW-HRP-V5-LRRTM2 (Addgene #82537) and FSW-TM-HRP (Addgene #82540). The DNA was amplified using the PureLink Expi Endotoxin-Free Maxi Plasmid Purification Kit (Thermo Fisher Scientific, A31231) and the lentiviral production was carried out by iVector facility of the Paris Brain Institute under BioSafety Level 2 (BSL-2) conditions. Briefly, HEK 293T cells were transfected with the plasmids of interest along with third generation packaging (transfer and envelope plasmids), where the vesicular stomatitis virus G glycoprotein (VSVG) was used as the envelope protein. To facilitate high transfection efficiency, lipofectamine 2000 (Merck, 11668500) was used in a medium containing chloroquine (Merck, 455245000). The HEK293T cells were maintained at 37˚C in a 95% air/5% $CO_2$ humidified incubator and 6 h after transfection the transfection reaction was finished by media replacement. After 36 h, the supernatant of the cell culture containing lentivirus was collected, treated with DNAse I and ultracentrifuged at 22,000 rpm for 90 min at 4˚C using a Beckman Coulter ultracentrifuge (rotor SW28). Then, the pellet was resuspended in 0.1 M PBS, aliquoted and frozen at −80˚C until use. The efficiency of lentivirus production was measured by ELISA using the p24 ZeptoMetrix kit (Merck, 0801111) and presented a titer ranging from $3.10 \times 10^8$ to $2.84 \times 10^9$ viral particles per µl.

## Isolation of endogenous surface protein levels in the synaptic cleft by proximity labeling during KCl depolarization

Proximity biotinylation experiments were performed at 37˚C using a CultureTemp Warming Plate (Merck, BAF370151000) for keeping the temperature of the experiment constant. Primary cortical neurons were transduced at DIV15 with 2 lentiviruses to express simultaneously both FSW-HRP-V5-LRRTM1 and FSW-HRP-V5-LRRTM2 using around $5.94 \times 10^8$ total viral particles per dish containing 4 million neurons. The live cell biotinylation assay was carried out at DIV19 using two 10 cm dishes per condition. For control conditions labeling (synaptic cleft at rest), the biotinylation reaction was triggered using 100 µM BxxP impermeant-biotin phenol (ApexBio Technology, A8012) and 1 mM $H_2O_2$ (Merck, H1009) in Tyrode's buffer (142.25 mM NaCl, 4 mM $CaCl_2$, 3 mM KCl, 1.25 mM $MgCl_2$, 0.5 mM $NaH_2PO_4$, 10 mM Glucose, 10 mM HEPES, 10 µM 6-cyano-7-nitroquinoxaline-2,3-dione (CNQX), and 50 µM D,L-2-amino-5-phospho-novaleric acid (AP5)) for 15 or 30 s. For isolating the synaptic cleft during activity, the same reaction was performed but using a Tyrode's solution containing 90 mM KCl and reducing NaCl to 55.25 mM to keep the osmolarity constant between conditions. After the biotinylation reaction was finished, the reaction was quenched with 3 subsequent washes using the quenching buffer described in Loh and colleagues [24] based on Tyrode's solution supplemented with 10 mM sodium azide (Merck, S2002), 10 mM sodium L-ascorbate (Merck, 11140), and 5 mM Trolox (Merck, 238813); also, in our case, 10 µM CNQX and 50 µM AP5 were added. Next, neurons were washed once with cold 1X DPBS (Thermo Fisher Scientific, 14190094) and scrapped in 400 µl of cold 1X DPBS (for 1 dish). A fraction of this neuronal mixture was taken to measure protein concentration through Pierce Rapid Gold BCA Protein Assay Kit (Thermo Fisher Scientific, A53225) and the rest was used to isolate the biotinylated synaptic cleft proteins following the steps specified in Loh and colleagues [24]. In this sense, each sample was pelleted by centrifugation at 3,000 g for 10 min at 4˚C and the pellet was resuspended in 100 µl of 1% SDS lysis buffer (1% SDS in 50 mM Tris-HCl (pH = 8.0) and supplemented with 1X protease inhibitor cocktail (from 100X; Merck, P8849), 1 mM phenylmethylsulfonyl fluoride (PMSF; Roche, 10837091001), 10 mM sodium azide, 10 mM sodium L-ascorbate, and 5 mM Trolox). Next, following the optimization carried out in Loh

and colleagues [24] to remove postsynaptic density proteins, the samples were boiled at 95˚C for 5 min. Then, 400 μl of 1.25X RIPA lysis buffer (150 mM NaCl, 0.2% SDS (Merck, L3771), 0.5% sodium deoxycholate (Merck, D6750), 1% Triton X-100 (Merck, X100) prepared in 50 mM Tris-HCl (pH = 8.0)) were added per sample. The lysates were incubated for 30 min at 4˚C in rotation and then were cleared by centrifugation at 16,000 g for 10 min at 4˚C.

## Characterization of endogenous biotinylated proteins by western blotting

For gel-based visualization of whole lysates (Figs 1F and S2), 20 μg of cell lysate was combined with NuPAGE LDS Sample Buffer 4X (Thermo Fisher Scientific, NP0007) supplemented with 40 mM dithiothreitol DTT (Merck, D9779), boiled for 5 min at 90˚C and run on an 8% SDS-PAGE gel. The gel was transferred to a 0.22 μm nitrocellulose membrane (Amersham Protran, G10600080) through Mini Trans-Blot Electrophoretic Transfer Cell (BioRad, 1703930) for 90 min at 100 V. The membranes were stained with Ponceau S (for 1 min in 0.1% w/v Ponceau S in 5% (v/v) acetic acid/water) to verify the quality of the transfer. Next, the membranes were destained with deionized water and blocked with 10% Milk (Merck, 70166) in TBS-T (0.2% Tween-20 in 1X Tris-buffered saline) for beta Actin detection or with 3% w/v BSA in TBS-T for Streptavidin-HRP at RT for 1 h. In the first case, the membrane was incubated with anti-beta Actin primary antibody (Thermo Fisher Scientific, PA5-85271, 1:5,000 dilution) at 4˚C overnight in gentle agitation. The next day, the membrane was washed with 4 times (10 min/wash) with TBS-T, probed with Goat Anti-Rabbit IgG (H+L)-HRP conjugate secondary antibody (BioRad, 1706515, 1:10,000 dilution), washed 4 times with TBS-T and then developed with Clarity ECL Western Blotting Substrate (BioRad, 1705060) using for imaging one ChemiDoc Touch Imaging System (BioRad, 1708370). To check the global biotinylation reaction, the membrane was incubated with Pierce High Sensitivity Streptavidin-HRP (Thermo Fisher Scientific, 21130, 1:5,000 dilution in 3% w/v BSA in TBS-T) at RT for 1 h, washed 4 times with 1X TBS-T and developed as outlined above.

For western blotting visualization of the biotinylated proteins on the synaptic cleft, the cell lysates were incubated with 25 to 40 μl of Pierce Streptavidin Magnetic Beads slurry (Thermo Fisher Scientific, 88817) at 4˚C overnight with gentle rotation (100 μg cell lysate for Synaptophysin1, vGlut1 or Synaptotagmin1; 250 to 300 μg for NPTX1 or ATG9 detection; or 300 μg for experiments using HRP-TM as biotinylating enzyme). The following day, the beads were washed as described in Loh and colleagues [24] with 1 ml RIPA lysis buffer (×2), 1 ml of 1 M KCl, 1 ml of 0.1 M Na$_2$CO$_3$, 1 ml of 2 M urea (prepared in 10 mM Tris-HCl (pH 8.0)), again with 1 ml RIPA lysis buffer (×2) and finally we did one wash with 1 ml DPBS (all the washes were performed with gentle rotation through Tube Revolver (Thermo Fisher Scientific, Model 88881002)). Next, the elution of the biotinylated proteins was carried out boiling the beads in 25 μl of 3× protein loading buffer supplemented with 20 mM DTT and 2 mM Biotin (Merck, B4501) for 10 min at 95˚C in gentle shaking. To remove the supernatant after each beads wash and for collecting the final eluate a DynaMag -2 (Thermo Fisher Scientific, 12321D) was used. Finally, the streptavidin eluate was run on an 8% or 10% SDS-PAGE gel and the membranes were transferred to a 0.22 μm nitrocellulose membrane for 90 min at 100 V. Subsequently, membranes were stained with Ponceau S, blocked with 10% Milk and incubated at 4˚C overnight with primary antibodies.

The primary antibodies and corresponding dilutions used in this study were (Table 2).

All primary antibodies were prepared in 10% milk and preincubated for 30 min at RT before being added to the membrane. The following day after washes with TBS-T and incubation with secondary antibodies (Goat Anti-Rabbit or Goat Anti-Mouse IgG (BioRad 1706516) 1:5,000 dilution in 10% milk); the membranes were imaged as described above but using Clarity Max ECL Western Blotting Substrate (BioRad, 1705062).

**Table 2. Primary antibodies and corresponding dilutions used in this study.**

| Antibody target | Species | Dilution (v/v) | Company |
|---|---|---|---|
| GluA1 | Mouse | 1:500 | NeuroMab, 75–327 |
| ATG9A | Rabbit | 1:1,000 | Abcam, 108338 |
| NPTX1 | Rabbit | 1:750 | Proteintech, 20656-1-AP |
| SNAP25 | Mouse | 1:3,000 | Synaptic Systems, 111111 |
| Synaptophysin 1 | Mouse | 1:2,000 | Synaptic Systems, 105011 |
| VGLUT 1 | Mouse | 1:2,000 | Synaptic Systems, 135011 |
| Synaptotagmin 1 | Rabbit | 1:1,000 | Synaptic Systems, 105103 |
| Rab3 | Mouse | 1:1,000 | Synaptic Systems, 107011 |
| PSD95 | Rabbit | 1:2,000 | Proteintech, 20665-1-AP |

An extended supplementary protocol can be found in the Supporting information (S1 Protocol) describing all steps necessary for reproducing our method for capturing activity-driven translocation using proximity labeling.

## Characterization of biotinylation activity by microscopy

To characterize the biotinylation activity of peroxidase constructs by fluorescence microscopy, primary cortical neurons (38,000 cells per coverslip) were transfected at DIV7 with either both FSW-HRP-V5-LRRTM1 and FSW-HRP-V5-LRRTM2, or with cytosolic APEX (NES-A-PEX2-GFP) constructs, using the calcium phosphate method described above. At DIV19, a live cell biotinylation assay was performed with 100 μM membrane-impermeant BxxP and 1 mM $H_2O_2$ in Tyrode's buffer, running the reaction for 15 s at 37°C. After this time, the reaction was quickly quenched with 3 continuous washes using the quenching buffer described by Loh and colleagues [24], supplemented with 10 mM sodium azide, 10 mM sodium ascorbate, and 5 mM Trolox.

Next, neurons were fixed with 4% paraformaldehyde at RT for 10 min and washed 3 times with 1× DPBS with gentle shaking. After fixation, they were permeabilized with Digitonin (Merck, D141; 0.01% solution in 3% w/v bovine serum albumin (BSA) in 1× DPBS) for 1 min at RT, followed by 2 washes with 1× DPBS, and blocked with 3% BSA in 1× DPBS for 1 h at RT. The neurons were then incubated with the primary antibodies: anti-V5 for neurons transfected with LRRTM constructs or anti-GFP for those transfected with Cyto-APEX. Both antibodies were prepared in blocking solution at a 1:1,000 dilution. This incubation was done overnight at 4°C in a humidified chamber with gentle agitation. The next day, the cells were washed 3 times with 1× DPBS with gentle agitation. They were then incubated with secondary antibodies (Goat anti-Mouse or anti-Rabbit IgG (H+L)-Alexa Fluor 488) and Streptavidin Alexa Fluor 647, both at a 1:1,000 dilution in 3% BSA in 1× DPBS, for 1 h at RT in a humidified chamber with gentle agitation in the dark. Note that this protocol permeabilizes neurons before biotin staining, enabling the detection of intracellular labeling if it had occurred when providing BxxP as a substrate. Finally, the cells were washed 3 times with 1× DPBS and visualized using a Zeiss Axio Observer 3 epifluorescence microscope.

Antibodies used for immunofluorescence experiments and their dilutions are listed below (Table 3).

## Live imaging of neurons

Live imaging assays of cotransfected primary cortical neurons were performed from DIV12 to DIV21. Imaging experiments were performed using a custom-built laser illuminated

**Table 3. Antibodies used for immunofluorescence experiments and their dilutions.**

| Antibody target | Species | Dilution (v/v) | Company |
|---|---|---|---|
| V5 | Mouse | 1:1,000 | Thermo Fisher Scientific, R960-25 |
| GFP | Rabbit | 1:1,000 | Thermo Fisher Scientific, A6455 |

epifluorescence microscope (Zeiss Axio Observer 3) coupled to an Andor iXon Life camera (model IXON-L-897), whose chip temperature is cooled down to −70˚C. Illumination using fiber-coupled lasers of wavelengths 488 (Coherent OBIS 488 nm LX 30 mW) and 561 (Coherent OBIS 561 nm LS 80 mW) was combined through using the Coherent Galaxy Beam Combiner, and laser illumination was controlled using a custom Arduino-based circuit coupling imaging and illumination. Neuron-astrocyte co-cultures grown on coverslips were mounted on a closed bath imaging chamber for field stimulation (Warner Instruments, RC-21BRFS) and imaged through a 40× Zeiss oil objective Plan-Neofluar with an NA of 1.30 (WD = 0.21 mm). Neurons were stimulated using field stimulation as indicated in each experiment, which was achieved by passing 1 ms current pulses between the platinum-iridium electrodes of the stimulating chamber (Warner Instruments, PH-2) that were induced using a stimulus isolator (WPI, MODEL A382) whose output was controlled by the same custom Arduino-based circuit coupling imaging and stimulation. Changes in fluorescence were captured with an imaging frequency of 5 Hz. For normalization of the signals, neurons were first exposed to a quenching solution at pH 5.5 buffered with 2-(N-morpholino)ethanesulfonic acid (MES) and then total fluorescence was revealed using a solution containing 50 mM $NH_4Cl$ (pH 7.4). These calibration experiments presented identical illumination intensities and exposure times but were acquired with an imaging frequency of 2 Hz.

All the experiments were performed at 36.5˚C. Temperature was kept constant using a Dual Channel Temperature Controller (Warner Instruments, TC-344C) that controlled the temperature of the stimulation chamber (Warner Instruments, PH-2) and simultaneously warmed the imaging solutions using an in-line solution heater (Warner Instruments, SHM-6). Imaging solutions were flown at 0.35 to 0.40 ml/min. Imaging was performed using a Tyrode's solution composed of (in mM) 119 NaCl, 2.5 KCl, 2 $CaCl_2$, 2 $MgCl_2$, 20 Glucose together with 10 μM CNQX and 50 μM AP5, buffered to pH 7.4 at 37˚C using 25 mM HEPES. Calibrating solutions containing $NH_4Cl$ at pH 7.4 were as the Tyrode's solution except they contained 50 mM $NH_4Cl$ and 69 mM NaCl. The composition of the surface quenching solution was as the Tyrode's solution but buffered with MES instead of HEPES in order to reach pH 5.5 at 37˚C.

## Imaging of pHluorin-tagged constructs

Quantitative measurements of exocytosis using pHluorin cannot be analyzed with standard $\Delta F/F_0$ calculations, as the resting $F_0$ cannot accurately reflect the expression levels of the sensor because the contribution of intracellularly localized pHluorin molecules will be equally negligible in different neurons presenting different levels of expression. Thus, we normalized pHluorin changes during trafficking to the total expression of pHluorin in each presynaptic arborization, which we obtained by application of a Tyrode's solution containing 50 mM $NH_4Cl$ at pH 7.4, providing the percentage of mobilized pHluorin molecules mobilized in respect to the total pool. Trafficking was induced by either field stimulation as described above or by application of a Tyrode's solution containing KCl as described in experiments for proximity biotinylation.

## Surface fraction estimates from pHluorin-tagged constructs

The intensity of pHluorin fluorescence at a particular time point depends on the fraction of pHluorin molecules existing in their deprotonated form, which is nonlinearly modulated by pH. To estimate the fraction of pHluorin molecules located at the surface or intracellularly within presynaptic sites, we measured fluorescence from quenched terminals at pH 5.5 and terminals in which total fluorescence is revealed by 50 mM $NH_4Cl$, which allowed us to obtain the range of minimum and maximum possible fluorescence in each neuron. Assuming that the pKa of pHluorin is 7.1, we followed established protocols for estimating the number of molecules at the surface and intracellularly [10]. To obtain activity-driven fold changes in surface abundance using pHluorin, we stimulated neurons with 200 AP fired at 50 Hz, and divided the calculated values of surface fraction obtained after electrical stimulation by the surface fraction values obtained at rest.

## Image analysis and statistics

For statistical analysis, GraphPad Prism v9 was used. Data sets were analyzed for normality using Shapiro–Wilk test, and depending on the results, appropriate parametric or nonparametric tests were applied, as indicated in S1 Table. Statistical significance was set at $p < 0.05$ and it is indicated by one asterisk, while $p < 0.01$ and $p < 0.001$ are indicated by 2 and 3 asterisks, respectively.

Imaging experiments using pHluorin constructs were replicated in independent experiments, as indicated in S1 Table. Images were analyzed using the ImageJ plugin Time Series Analyzer V3 by selecting 150 to 200 regions of interest (ROIs). These ROIs correspond to synaptic boutons that showed changes in response to 50 mM $NH_4Cl$. As such, we obtained fluorescence changes over time from all boutons of entire presynaptic arborizations, unless otherwise noted.

For western blot quantification, we used Image Lab software (BioRad, Version 6.1.0 build 7) to analyze the total volume of bands of interest and subtract the volume of a membrane space where no bands were present (background). All results were normalized by the value obtained in GluRI, unless otherwise noted. Due to the variability of raw intensities between western blots from different experiments, we studied relative changes in band intensity by normalizing control values. One sample Wilcoxon Signed-Rank tests were used to compare whether the median of the normalized values significantly differed from 1, representing the control. The correlation between fold changes in synaptic surface measured using pHluorin and synaptic cleft isolation was assessed using simple linear regression. The analysis included calculating the 90% confidence bands of the best-fit line, with the fit constrained to pass through the origin, as no change indicates no translocation for both pHluorin or cleft isolation. Deviation from linearity was assessed using the runs test.

## Ontology analysis using SynGO

Genes corresponding to the excitatory synaptic cleft proteome were downloaded from published data sets generated by Loh and colleagues [24]. The published excitatory synaptic cleft proteome consisted of 199 rat uniprot IDs, which can be obtained from the Table S1 of Loh and colleagues [24]. We converted these into human gene IDs using the SynGO ID convert tool (https://www.syngoportal.org/convert). The gene data set corresponding to the excitatory synaptic cleft was then intersected with the list of SynGO1.2 annotated genes at www.syngoportal.org. Not all genes identified by Loh and colleagues [24] were present in the SynGO database at the time of the analysis. From the data set, 98 genes presented a Cellular Component annotation (i.e., location annotation). This data set was used to provide a

comparison to a defined background set of brain expressed genes using a Fisher exact test as described by Koopmans and colleagues [29]. The brain expressed background set selected contains 18,035 unique genes in total, of which 1,591 overlap with SynGO1.2 annotated genes. Using this analysis, we found that 18 Cellular Component terms (i.e., location annotations) were significantly enriched at a 1% false discovery rate (FDR). We tested terms with at least 3 matching input genes and generated a figure in which each possible subcellular location annotation predefined in SynGO1.2 was color-coded to reflect the statistical significance of the multiple comparisons, shown as the $-\log_{10}$ of the Q-value. Subcellular localizations that were not represented by at least 3 input genes were designated as "too few genes" and were colored in light gray. To highlight that synaptic vesicle localization terms were not identified in the synaptic cleft data set, we colored them in green and annotated them as "no SV genes found." The specific synaptic vesicle location terms that failed to be identified from the genes contained in the synaptic cleft proteome follow a hierarchical tree structure from broader to more specific terms as follows: synaptic vesicle > synaptic vesicle membrane > integral/extrinsic/anchored component of synaptic vesicle membrane, as described by Koopmans and colleagues [29].

## Supporting information

**S1 Fig. LRRTM-HRP expression does not alter SV properties. (A)** Gene ontology analysis of published datasets of the excitatory synaptic cleft using SynGO shows the subcellular location of identified genes, showing the corresponding $-\log_{10}$ of the Q-value of genes significantly enriched. We tested terms with at least 3 matching input genes. Subcellular localizations that were not represented by at least 3 input genes were designated as "too few genes." No genes were associated with the SynGO term Synaptic vesicle, highlighted in green. Presynaptic terms are under the label "Pre," postsynaptic terms under the label "Post." **(B)** Traces of the different experiments performed for Fig 1B show synaptic vesicle exocytosis using synaptophysin-pHluorin (Syp-pH) at different times (15, 30, and 60 s) of KCl treatment. The traces are normalized to the peak response. **(C)** Synaptic vesicle pool size was measured by revealing the total fluorescence of Syp-pH with $NH_4Cl$ at pH 7.4 in neurons that were or not transduced with lentiviruses to express LRRMT-HRP (denoted as +HRP in red), showing no difference in SV pool size. S1 Table shows the number of experiments and replicates, means and error, and statistical tests used. (EPS)

**S2 Fig. Surface biotinylation using BxxP and KCl does not alter labeling rates. (A)** Global biotinylation rates measured from total lysates using streptavidin-HRP in neurons expressing LRRTM-HRP under control (C) or depolarization (KCl) conditions tested for 15 s. **(B)** Quantification of experiments shown in (A). **(C)** Synaptic cleft biotinylation rates measured from enriched biotinylated proteins using streptavidin-HRP in neurons expressing LRRTM-HRP under control (C) or depolarization (KCl) conditions tested for 15 s. **(D)** Quantification of experiments shown in (C). **(E)** Neurons transfected to express surface LRRTM-HRP or cytosolic APEX (NES-APEX2-GFP) were subjected to a biotinylation reaction using the impermeant BxxP biotin substrate. Note that no biotinylation signal is found intracellularly when using cytosolic APEX, yet a clear signal is found if HRP is located at the surface of neurons. Staining anti-V5 or anti-GFP shows the expression of the biotinylating enzyme in each case. Scale bar = 20 μm. S1 Table shows the number of experiments and replicates, means and error, and statistical tests used. (EPS)

**S3 Fig. NPTX1 and ATG9A present high intracellular fraction at presynapses. (A)** Representative images of presynaptic arborizations expressing both Synapsin-mRuby (Presynapse)

and NPTX1-pH. Baseline fluorescence appears to be relatively low and application of an impermeant quenching solution at acidic pH (MES pH 5.5) reduces fluorescence levels slightly. Revealing the total expression of NPTX1-pH at the presynapse using $NH_4Cl$, which de-acidifies internal compartments, shows high fluorescence changes, indicating that a large fraction of NPTX1-pH is located in acidic intracellular compartments. Scale bar = 4.8 μm. **(B)** Quantification of the percentage of NPTX1-pH molecules located intracellularly in presynaptic arborizations of different neurons. **(C)** Quantification of NPTX1-pH exocytosis in individual neurons in response to 15 s of KCl application, shown in gray. Green trace shows the average response. Traces are normalized to the total NPTX1-pH mobilizable pool, obtained by application of $NH_4Cl$ (pH 7.4). **(D)** Representative images of presynaptic arborizations expressing both Synapsin-mRuby (Presynapse) and ATG9A-pH. Images show the same conditions as in (A), revealing that a large fraction of ATG9A-pH is located in intracellular acidic compartments. Scale bar = 4.8 μm. **(E)** Quantification of the percentage of ATG9A-pH molecules located intracellularly in presynaptic arborizations of different neurons. **(F)** Quantification of ATG9A-pH exocytosis in individual neurons in response to 15 s of KCl application, shown in gray. Green trace shows the average response. Traces are normalized to the total ATG9A-pH mobilizable pool, obtained by application of $NH_4Cl$ (pH 7.4). **(G)** Quantification of surface abundance changes in ATG9A, NPTX1, vGlut1, Syp1, and Syt1 by synaptic cleft isolation and pHluorin imaging. Changes in pHluorin abundance are quantified by measuring relative changes in surface fraction induced KCl application for 15 s, while changes in the synaptic cleft of endogenous proteins are measured as in Fig 4F. A simple linear regression is shown (continuous gray line) along with its 90% confidence bands (dotted lines). S1 Table shows the number of experiments and replicates, means and error, and statistical tests used. (EPS)

**S1 Table. Description of the statistics used in the present study.** Statistics table reporting information on the number of experiments and biological replicates, means and error, and different statistical tests used.
(XLSX)

**S1 Data. Summary of data used to represent the results of the study.** This data collection contains the absolute and raw values for representing the results shown in Figs 1C, 1D, 1E, 2B, 2C, 2E, 3B, 3E, 4B, 4E, 4F, 4G, 4H, S1B, S1C, S2B, S2D, S3B, S3C, S3E, S3F, and S3G.
(XLSX)

**S1 Raw Images. Original images of western blotting experiments.** This file contains the uncropped chemiluminescence images for the western blot results represented in Figs 1F, 2A, 2D, 3A, 4A, S2A, and S2C.
(PDF)

**S1 Protocol. Extended protocol for synaptic cleft proximity labeling during neuronal activity.** This section contains the step by step detailed protocol for the preparation of neuronal embryonic cultures, synaptic cleft proximity biotinylation and lysate collection for some of the experiments carried out in this study.
(DOCX)

## Acknowledgments

We thank Franck T. Koopmans (Vrije Universiteit Amsterdam) for insightful comments on the manuscript and Kahina Boumendil (Paris Brain Institute) for excellent technical assistance.

## Author Contributions

**Conceptualization:** Carlos Pascual-Caro, Jaime de Juan-Sanz.

**Funding acquisition:** Carlos Pascual-Caro, Jaime de Juan-Sanz.

**Investigation:** Carlos Pascual-Caro.

**Methodology:** Carlos Pascual-Caro, Jaime de Juan-Sanz.

**Project administration:** Jaime de Juan-Sanz.

**Supervision:** Jaime de Juan-Sanz.

**Writing – original draft:** Carlos Pascual-Caro, Jaime de Juan-Sanz.

**Writing – review & editing:** Carlos Pascual-Caro, Jaime de Juan-Sanz.

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
