## [Editor Report · Decision Letter 0]

3 Apr 2024

Dear Dr de Juan-Sanz, 

Thank you for submitting your manuscript entitled "Activity-driven trafficking of endogenous synaptic proteins through proximity labeling" for consideration as a Methods and Resources Article by PLOS Biology. Please accept my apologies for the delay in getting back to you as we consulted with an academic editor about your submission. Please note that I am currently handling your manuscript since my colleague Christian Schnell, who is the handling editor for your submission, is now away from the office this week. 

Your manuscript has now been evaluated by the PLOS Biology editorial staff, as well as by an academic editor with relevant expertise, and I am writing to let you know that we would like to send your submission out for external peer review.

Once your full submission is complete, your paper will undergo a series of checks in preparation for peer review. After your manuscript has passed the checks it will be sent out for review. To provide the metadata for your submission, please Login to Editorial Manager (https://www.editorialmanager.com/pbiology) within two working days, i.e. by Apr 05 2024 11:59PM.

Kind regards,

Richard 

Richard Hodge, PhD

rhodge@plos.org

On behalf of:

Christian Schnell, PhD

cschnell@plos.org

PLOS

---

## [Decision Letter · Decision Letter 1]

21 May 2024

Dear Dr de Juan-Sanz,

Thank you for your patience while your manuscript "Activity-driven trafficking of endogenous synaptic proteins through proximity labeling" was peer-reviewed at PLOS Biology. It has now been evaluated by the PLOS Biology editors, an Academic Editor with relevant expertise, and by several independent reviewers. 

In light of the reviews, which you will find at the end of this email, we would like to invite you to revise the work to thoroughly address the reviewers' reports.

As you will see below, the reviewers think that the study provides important insights. They mostly ask for additional control experiments, additional methodological details but also request a few more experiments. While we think that these would all provide valuable additional insights, we don't think that requests #3 and #4 are strictly necessary in the context of this revision. 

Given the extent of revision needed, we cannot make a decision about publication until we have seen the revised manuscript and your response to the reviewers' comments. Your revised manuscript is likely to be sent for further evaluation by all or a subset of the reviewers.

**IMPORTANT - SUBMITTING YOUR REVISION**

*Re-submission Checklist*

*Published Peer Review*

*PLOS Data Policy*

*Blot and Gel Data Policy*

Sincerely,

Christian

Christian Schnell, PhD

Senior Editor

PLOS Biology

cschnell@plos.org

REVIEWS:

Reviewer #1: General assessment:

The goal of the authors is to develop a method that identifies the activity-driven synaptic cleft proteome, an important and outstanding question in the field. This work builds on a previously developed tool that localized the biotinylating agent to the postsynaptic membrane, but that, as used, could not identify the proteins that are transiently localized to the cleft during activity-driven SV exocytosis. Through spatiotemporal coincidence of neuronal activity and biotinylation, the authors show that they can detect proteins that could not be detected with the existing approaches. They validate the method by looking at two well-known synaptic vesicle proteins, and then extent their technique to demonstrate that ATG9 and NPTX1, two proteins that have been hypothesized to be present at the cleft during activity-driven exocytosis, could be detect through the newly developed biotinylation assays and through traditional pHluorin tagging. The technique is of value and beyond the technique, the studies on ATG9 and NPTX1 are of value to the field. We suggest additional controls and minor edits to improve the paper.

Major revisions:

1. The results for one of these two positive controls, VGlut1, is confusing. Can the authors comment on the high levels of vGlut1 in the Control lane under the 30s condition? Is this consistently observed? To that avail, could there be additional positive controls, like synaptotagmin proteins, that might strengthen these control observations?

2. It is not clear how the normalizations were performed for Fig 2B, C and similar graphs. Is it well known (and expected) that GluR1 does not vary due to activity at the synapse? 

3. What explains the higher variability observed with the 15s KCl incubation? In the supplement it would be ideal to add curves of the mean+error bars of all the experiments that appear in 1C. In Figure 2B-C, the spread is also high in 15s, higher than that observed in 30s conditions. In general, it would, be beneficial for the authors to comment on this variability observed for the technique, and how the variability observed for other biotinylation approaches.

4. Because the temporal aspect of the biotinylation and activation are so crucial for this approach, in the results, the authors should briefly explain how to controlling biotynilation times, and extend this part o the discussion to future developments that would help better identify the activity-driven synaptic cleft proteome.

Minor revisions:

5. Provide an explanation as to why imaging experiments were done in postnatal cortical neurons while the biotinylation assay was done in embryonic ones. 

6. Where they Reframe Figure S1A - it is not clear what each square represents and it is not explained in the methods which datasets they used to create this figure or how they did it. What do they mean by "too few SV genes"?

7. Figure 1A might benefit from having a legend that defines what each color or shape represents.

8. All plots that have error bars should specify which type of error bars they are. SD or SEM?

9. In Figure C, what does each dot represent? Additionally, please place the statistical comparisons closer to the pair of groups that are being compared. Did the authors do a comparison between 15 and 30 seconds? 

Reviewer #2: The manuscript "Activity-driven trafficking of endogenous synaptic proteins through proximity labeling" developed an approach using synaptic cleft proximity labeling to capture and quantify activity driven trafficking of endogenous synaptic proteins at the synapse. The manuscript claimed that by carefully coordinating biotinylation and depolarization at synaptic clefts, and reducing labeling times to limit nonspecific labeling, proximity labeling enables the specific isolation and quantification of endogenous translocated proteins to the synaptic surface during neuronal activity. The main claim of the manuscript suggests the potential of this approach in providing direct evidence of the surface translocation of non-canonical trafficking proteins, such as ATG9A and NPTX1. The study adds to growing evidence that proximity labeling has emerged as one of the most useful research technologies to explore the trafficking of endogenous non-canonical proteins at the presynapse. The manuscript claimed that this strategy can be used to capture the translocation of any possible endogenous synaptic protein during neuronal activity. Further study demonstrated that endogenous NPTX1 and ATG9A are translocated to the synaptic surface, providing important biological insight into the functioning of these proteins at the synapses. 

However, I have several major concerns that it needs more solid evidence to prove the author's viewpoint for publication. While the evidence that the approach enables capturing endogenous synaptic proteins involving into activity-driven trafficking through proximity labeling, the efficiency of this approach as well as the advantage of this approach need to be determined. It was also determined whether this approach can be used to identify new endogenous synaptic proteins in the synaptic surface during neuronal activity. Although this is potentially an important approach emphasizing the application of proximity labeling in the study of endogenous synaptic proteins, several data sets are preliminary.

Notwithstanding the above, I do have some comments/suggestions for the author's consideration. As listed below.

1. The authors claimed that this approach facilitates directly assessing which certain proteins, hypothesized to participate in activity-driven translocation at synapses, indeed engage in such a process. However, the existed pHluorin-tagged constructs for these proteins also can enable such purpose to be achieved. From Figure 4F-4H, pHluorin-tagged constructs showed stronger signal intensity than the authors' approach. For example, pHluosin-vGlut1 showed a ~6-fold increase in the number of molecules found in surface abundance after field stimulation of 200 AP at 50 Hz, whereas vGlut1 showed only ca 3-fold increase in western blot of the isolation of synaptic cleft in the presence of KCl. pHluosin imaging methods also display advantage for other trafficking proteins, such as Syp1 and ATG9A. The authors should perform two methods, pHluosin imaging and synaptic cleft isolation, in same experimental conditions. Also, the authors did explain why field stimulation, but not KCL, was used to stimulate cells.

2. The authors claimed that the approach revealed all biotinylated proteins, which appeared as a smeared band spanning most molecular weights in cells expressing LRRTM-HRP construct (Figure 1F), and quantification of total biotinylation rates revealed no significant differences (Supp. Fig. S1C, D). Theoretically, KCl treatment causes more proteins translocated to synaptic cleft. The authors should provide evidence to show western blot of all biotinylated proteins from the isolation of synaptic cleft in the presence of KCl. 

3. In addition, the authors should perform LC-MS/MS to identify all biotinylated proteins from the isolation of synaptic cleft in the presence of KCl. Proteomic mapping of endogenous synaptic proteins in activity-driven trafficking would be a very useful resource. Please refer to Ting's work (Cell. 2016 Aug 25;166(5):1295-1307).

4. In this manuscript, the authors checked synaptic proteins during depolarization with KCl. I would expand the approach in excitatory synapses or inhibitory synapses. 

5. In this approach, coordinating proximity labeling times and translocation dynamics is essential to detect relative changes in surface abundance of trafficking proteins. The authors claimed that 15s is optimal biotinylating time and translocation time. Proteins from different trafficking pathways vary, and they take different time to reach plasma membrane. So it is very different to capture most of endogenous translocation proteins in the complicated context of cellular biology.

Minor comments:

1. Page 2, 3rd paragraph: HPR should be HRP.

2. Page 2, 3rd paragraph: H202 should be H2O2.

3. Page 21, the legend of Figure 1: "potassium (KCl)" should be "potassium chloride (KCl)".

4. Page 22, Figure 2A: the WB picture was cropped unproperly. Please provide full picture for the Syp1 in the Input. 

5. Glutamate Receptor 1 should be abbreviated as "GluR1" (Figure 2B and 2C) or "GluR I" (in the text)? 

6. Please label the scale bar in Figure 3D. 

Reviewer #3: Pascual-Claro and de Juan-Sanz report a new approach and use of the repurposed technology to evaluate the transient exposure of synaptic vesicle (SV) proteins to the extracellular milieu. The authors built on the proximity labelling approach coupled with elicited increased synaptic activity to promote exocytosis of numerous SVs, thus increasing the abundance of the (integral) SV proteins at the synaptic cleft. The use of the fast kinetics of the HRP enzyme allowed the authors to study the trafficking of endogenous proteins to the plasma membrane of the synapse. Although similar approaches to study these questions have been documented in the literature (as noted by the authors too), this field would benefit from an unbiased proteomic-based approach to identify proteins present in the synaptic membranes shortly after stimulation. 

Yet the novel scientific outcomes gained from this technology are somewhat underwhelming, based on the presented examples of the neuronal pentraxin 1 (a soluble, secreted glycoprotein) and Autophagy-Related 9A (ATG9A). Given that the findings are presented through the methodology-driven manuscript, it is not clear if the identification of labelled proteins is still undergoing. Either way, there are several points that should receive further attention:

1. The authors state that they used LRRTM1-HRP and LRRTM2-HRP to perform proximity labelling. However, it is not clear which construct is used in which experimental setup. This should be clarified in the manuscript.

2. It may be useful to detail how synaptic cleft isolation is performed here. Do the results reveal biotinylated proteins present at the pre-synapse after de-enrichment of post-synaptic densities? The list of biotinylated proteins after stimulation (or pre/post stimulation difference) by mass spectrometry, and verified by Western blotting for key examples, would be useful.

3. Although LRRTM-HRP is expressed and only active when H2O2 and biotin are added (which should label only the surface-exposed proteins), the enzyme can be activated intracellularly at the time when H202 is added to the culture. Although the authors used an impermeant version of biotin, biotin is usually present in the culture media and can be available to the HRP ligase that is being trafficked intracellularly. Is any sort of biotin starvation performed prior to the KCl/biotin administration? Also, is there any biotinylation in the post-synaptic cell during the 15s stimulation? This can be addressed by biotin staining and/or by evaluating the presence of intracellular post-synaptic proteins after biotin pull-down. Specifically, Fig 2. panel A: is PSD-95 present in the isolated fractions? The authors should include the Western blot showing enrichment for biotin.

4. Fig 2.: SNAP-25 may not be ideal control in this context - it may be good to include other/additional control protein.

5. Fig 2. panel D: usually very little vGlut1 is present at the neuronal surface under the resting conditions (here control/C). It is surprising that much vGlut1 is detected after total surface isolation of biotinylated proteins. This control experiment, and the use of vGlut1 as an example here, is confusing and may not be ideal to convey the point that the authors wish to make.

6. Elevated extracellular potassium chloride is commonly used to achieve membrane depolarization of cultured neurons, yet it does not always induce neuronal activity and it is considered to be rather poor representation of in vivo phenomena. Does electrical stimulation of cultured neurons yield the same results?

7. Given the nice and inclusive introduction, it may be good to also cite the work presented in 10.1016/j.cell.2023.05.044

8. I appreciate the indicated (minor) gap in knowledge regarding neuronal pentraxin 1, yet it may be useful to elaborate why, out of all possible options and given limitations of this technology, the attention has been drawn to a soluble secreted protein. Are many other soluble proteins labelled too?

---

## [Decision Letter · Decision Letter 2]

5 Sep 2024

Dear Jaime,

Thank you for your patience while we considered your revised manuscript "Activity-driven trafficking of endogenous synaptic proteins through proximity labeling" for consideration as a Methods and Resources at PLOS Biology. Your revised study has now been evaluated by the PLOS Biology editors, the Academic Editor and two of the original reviewers. 

In light of the reviews, which you will find at the end of this email, we are pleased to offer you the opportunity to address the remaining points from the reviewers in a revision that we anticipate should not take you very long. We will then assess your revised manuscript and your response to the reviewers' comments with our Academic Editor aiming to avoid further rounds of peer-review, although might need to consult with the reviewers, depending on the nature of the revisions.

Please also make sure to address the following data and other policy-related requests:

* We would like to suggest a different title to improve readability/accuracy: "Monitoring of activity-driven trafficking of endogenous synaptic proteins through proximity labeling"

* Please add the links to the funding agencies in the Financial Disclosure statement in the manuscript details

* Please include the approval/license number of the ethical approval for the animal experiments.

* DATA POLICY:

Regardless of the method selected, please ensure that you provide the individual numerical values that underlie the summary data displayed in the following figure panels as they are essential for readers to assess your analysis and to reproduce it: 1CEF, 2BCE, 3B, 4FGH, S1C, S2BD and S3BEG

* CODE POLICY

* Please note that per journal policy, the model system/species studied should be clearly stated in the abstract of your manuscript. 

**IMPORTANT - SUBMITTING YOUR REVISION**

*Resubmission Checklist*

*Published Peer Review*

*PLOS Data Policy*

*Blot and Gel Data Policy*

Sincerely,

Christian

Christian Schnell, PhD

Senior Editor

PLOS Biology

cschnell@plos.org

REVIEWS:

Reviewer #1: We thank the authors for addressing most of our concerns. Our only remaining commentary regards the Supplementary Figure 1. We have two comments:

1. There seems to be some typos and mismatches between figure correspondence in the Table and the Figures of the paper, including references to statistically significant differences in the graphs of the main Figure, which do not apear as significant in the table (eg comparison between 15 and 30s has a p-value of 0.086 that is reported as significant in the main Figure). Adding p-values to the main figure legends will help this too.

2. We also have a question about the statistical methods used. The ratios, proportions or percentages values analyzed by the authors correspond to categorical variables, not numerical values. If so, they should be compared using a Chi-square test rather than a t-test or ANOVA test. 

Reviewer #3: I appreciate the efforts that authors put in the revision of original manuscripts, yet those efforts were predominantly focused on clarifications of the original work and adding additional controls. I was not aware of authors recent Cell Reports paper when originally reviewing this work (not published at that time point, I think). The suggested experiment that authors consider to be outside the scope of this work would, in my view, make this publication stronger and add to the novelty. 

This manuscript contains a nice set of data, which is now made even stronger through the revised work, yet the novelty is still limited. A good opportunity has possibly been missed here. 

That said, I support publication of dataset presented here.

---

## [Editor Report · Decision Letter 3]

20 Sep 2024

Dear Jaime,

Thank you for the submission of your revised Methods and Resources "Monitoring of activity-driven trafficking of endogenous synaptic proteins through proximity labeling" for publication in PLOS Biology. On behalf of my colleagues and the Academic Editor, Jonathan Demb, I am pleased to say that we can in principle accept your manuscript for publication, provided you address any remaining formatting and reporting issues. These will be detailed in an email you should receive within 2-3 business days from our colleagues in the journal operations team; no action is required from you until then. Please note that we will not be able to formally accept your manuscript and schedule it for publication until you have completed any requested changes.

When you attend to the requests to come, please also make sure that you reference the correct source data file in the figure legends.

PRESS

We frequently collaborate with press offices. If your institution or institutions have a press office, please notify them about your upcoming paper at this point, to enable them to help maximize its impact. If the press office is planning to promote your findings, we would be grateful if they could coordinate with biologypress@plos.org. If you have previously opted in to the early version process, we ask that you notify us immediately of any press plans so that we may opt out on your behalf.

Sincerely, 

Christian

Christian Schnell, PhD

Senior Editor

PLOS Biology

cschnell@plos.org